# *Lactobacillus casei* Improve Anti-Tuberculosis Drugs-Induced Intestinal Adverse Reactions in Rat by Modulating Gut Microbiota and Short-Chain Fatty Acids

**DOI:** 10.3390/nu14081668

**Published:** 2022-04-17

**Authors:** Yue Li, Liangjie Zhao, Meiling Hou, Tianlin Gao, Jin Sun, Hao Luo, Fengdan Wang, Feng Zhong, Aiguo Ma, Jing Cai

**Affiliations:** 1Department of Nutrition and Food Hygiene, School of Public Health, Qingdao University, Qingdao 266021, China; liy_moon@163.com (Y.L.); zhaoliangjie17@163.com (L.Z.); hougreensky@163.com (M.H.); gaotl@qdu.edu.cn (T.G.); sunj@qdu.edu.cn (J.S.); luo2186712@163.com (H.L.); sdytzywfd@163.com (F.W.); zhfeng@qdu.edu.cn (F.Z.); magfood@qdu.edu.cn (A.M.); 2Institute of Nutrition and Health, School of Public Health, Qingdao University, Qingdao 266021, China

**Keywords:** anti-TB drug, *Lactobacillus casei*, gut microbiota, short chain fatty acid

## Abstract

The adverse effects of anti-tuberculosis (TB) drugs in the intestines were related to alteration of the intestinal microbiota. However, there was less information about microbial metabolism on the adverse reactions. This study aimed to explore whether *Lactobacillus casei* could regulate gut microbiota or short-chain fatty acids (SCFAs) disorders to protect intestinal adverse reactions induced by isoniazid (H) and rifampicin (R). Male Wistar rats were given low and high doses of *Lactobacillus casei* two hours before daily administration of anti-TB drugs. After 42 days, colon tissue and blood were collected for analysis. The feces at two-week and six-week were collected to analyze the microbial composition and the content of SCFAs in colon contents was determined. Supplementation of *Lactobacillus casei* increased the proportion of intestinal goblet cells induced by H and R (*p* < 0.05). In addition, HR also reduced the level of mucin-2 (*p* < 0.05), and supplementation of *Lactobacillus casei* restored. After two weeks of HR intervention, a decrease in OTUs, diversity index, the abundance of *Bacteroides*, *Akkermansia*, and *Blautia*, and an increase of the abundance of *Lacetospiraceae NK4A136 group* and *Rumencoccus UCG-005*, were observed compared with the control group (*p* all < 0.05). These indices in *Lactobacillus casei* intervention groups were similar to the HR group. Six-week intervention resulted in a dramatic reduction of *Lacetospiraceae NK4A136 group*, butyric acid, valeric acid and hexanoic acid, while an increase of *Bacteroides* and *Blautia* (*p* all < 0.05). Pretreatment with *Lactobacillus casei* significantly increased the content of hexanoic acid compared with HR group (*p* < 0.05). *Lactobacillus casei* might prevent intestinal injury induced by anti-tuberculosis drugs by regulating gut microbiota and SCFAs metabolism.

## 1. Introduction

During tuberculosis (TB) treatment, isoniazid (H) and rifampicin (R), as the most widely used anti-TB drugs, can specifically identify and effectively kill *Mycobacterium tuberculosis* [1]. However, these drugs as antibiotics may cause gastrointestinal adverse reactions, which are the common adverse reactions during anti-TB treatment, including nausea, vomiting, dyspepsia, loss of appetite, diarrhea, and other symptoms, with an incidence of 3.45–29.4% [2,3,4,5]. The mechanism of antibiotic-induced adverse gastrointestinal symptoms mainly involves the destruction of intestinal microbiota [6]. Previously, studies also reported that the destruction of intestinal microbes induced by antibiotics may cause changes in the mucous layer, metabolome, and immune response [7,8]. Short-chain fatty acids (SCFAs) are produced by microorganisms in the large intestine during fermentation. SCFAs are the main anion in feces, which can maintain intestinal osmotic pressure and store energy. Antibiotic inhibition of the synthesis of short-chain fatty acids can lead to diarrhea [9]. In Namasivayam’s research, it was shown that anti-TB drugs could induce a distinct and long lasting dysbiosis and many of the affected taxa were associated with immunologic function [10], but the effect of anti-TB drugs on short-chain fatty acids, mucus layer and other intestinal barriers remains unclear.

Probiotics are defined as live microorganisms that confer a health benefit when consumed in adequate amounts [11]. Probiotics are used to cure children’s acute diarrhea [12] or antibiotic diarrhea [13]. As a common probiotic, *Lactobacillus casei* (*L. casei*) was reported to regulate intestinal microbiota [14] and relieve medical gastrointestinal discomfort caused by academic stress [15]. *L. casei ATCC334* also has anti-inflammatory properties which can reduce the abundance of pro-inflammatory cytokines in the zebrafish intestine [16]. The recent study on aged mice found that *Lactobacillus casei* combined with dietary fiber complex had a protective effect of gut barrier function, and increased SCFA concentration and gene expression of SCFA receptors in the small intestine [17]. Meanwhile, one study certified that the most apparent effect of *Lactobacillus casei* supplementation was the relative increase in SCFAs in antibiotic-treated rats, especially propionic acid and butyric acid [18].

Based on the above evidence, anti-TB drugs and probiotics may directly or indirectly affect the intestinal barriers by changing the intestinal microbiota. *L. casei* beverages have been shown to reduce the incidence of total gastrointestinal adverse events associated with anti-TB drugs in our previous clinical trial [19]. Whether probiotics are effective in alleviating intestinal microbes dysbiosis or intestinal barrier disturbance induced by anti-TB drugs has not yet been reported. Furthermore, there was not enough evidence of the association between SCFAs and intestinal microbes on the intestinal adverse reactions reduced by anti-tuberculosis drugs. The purpose of this study was to illustrate the effect of anti-tuberculosis drugs on the intestinal barrier and the protective effect of *Lactobacillus casei*, and to explore the potential relationship with SCFAs in intestinal adverse reactions caused by anti-tuberculosis drugs.

## 2. Materials and Methods

### 2.1. Animals

Male Wistar rats (seven-week-old, 230–250 g) were purchased from Pengyue Experimental Animal Breeding Co., LTD. (Jinan, China) and kept in the animal laboratory of Institute of Nutrition and Health of Qingdao University. In order to reduce the individualization of intestinal microbiota, the rats were given adaptive feeding for two weeks without restriction of water and food. Maintenance feed for laboratory rats was produced by Synergy Pharmaceutical Bioengineering Co., Ltd. (production batch number 20190424, Nanjing, China), and the specific ingredients were shown in Appendix A. All experimental conditions and operations were in accordance with the requirements of the Laboratory Animal Management and Use Guidelines. This experiment was approved by the animal experiment review committee of the animal research center of Qingdao University (ethical approval number QYFYWZLL26005).

### 2.2. Reagent Preparation

Sodium carboxymethyl cellulose (CMC) is the hydroxyl of cellulose ether after the generation of derivatives, safe and non-toxic, which has been used as a thickener or carrier for food or medicine [20]. It was previously used as a solvent in animal experiments [21]. In this study, suspension of the anti-TB drugs isoniazid (INH, Sigma-Aldrich Co., St. Louis, MO, USA) and rifampicin (RIF, Tokyo Chemical Industries Co., Ltd., Tokyo, Japan) was prepared by dissolving 0.5% CMC.

*L. casei ATCC334* was obtained from Chiba Biotechnology Co., Ltd., (QYC-2019011603, Xi’an, China), and had a bacterial content of 10^10^ CFU/g (double plate counting). The *L. casei ATCC334* strains produced by the company came from the Chinese industrial microorganism species protection and management center, and then the powder was obtained by vacuum freeze-drying technology *L. casei ATCC334* powder was stored in the refrigerator at 4 °C. To restore activity, *L. casei ATCC334* powder was dissolved with 0.9% normal saline before use. The number of viable bacteria in *L. casei ATCC334* powder was counted by the pour plate method, and the result was 8.7 × 10^9^ CFU/g. The intervention dose of *L. casei ATCC334* was converted to this dose.

### 2.3. Grouping and Modeling

After an acclimatization period of 2 weeks, forty rats were randomly divided into four equal groups, each consisting of 10 rats as follows: control group (CN) was administered intragastrically with normal saline solution (NSS) and 0.5% CMC at an interval of 2 h; model group (HR) was gavaged with INH (50 mg/kg·bw) and RIF (100 mg/kg·bw) after NSS for 2 h; low-dosage and high-dosage *L. casei ATCC334* (LLc and HLc) pretreatment groups were administered intragastrically with different doses of *L. casei ATCC334* (1.0 × 10^9^ or 2.0 × 10^9^ CFU/kg·day) two hours before INH and RIF challenge. The intervention lasted 42 days.

### 2.4. Sample Collection

During the intervention period, the body weight and food intake were weighed every week. Feces were collected as 3–4 pieces from each rat aseptically for 14 days and 42 days of intervention, and quickly put in −80 °C refrigerator. After 42 days, all rats were euthanized with 10% chloral hydrate (300 mg/kg·bw, intraperitoneal injection) after fasting for 12 h. After collecting abdominal aortic blood, the serum was prepared by centrifuged at 3000 rpm for ten minutes and then extracted the supernatant. One centimeter of ascending colon was taken, rinsed with NSS and preserved in 4% formalin solution. A total of 4 cm of distal colon was taken, collected mucus and intestinal tissue, respectively. In addition, the intestinal tissue was homogenized for index detection. The serum samples and the contents of the colon were stored at −80 °C.

### 2.5. Intestinal Histopathology

Distal colon tissue was fixed with 4% paraformaldehyde for 48 h. After gradient dehydration, samples were cut into 5 μm thick serial sections for hematoxylin and eosin (HE) staining. The sections were observed under a microscope (Olympus Corporation, Tokyo, Japan) to distinguish the morphological structure of normal tissue and pathological colon tissue. Under the 50× field of vision the samples could be found. Photos were taken at 200× field of vision for counting goblet cell. The ratio of goblet cells was defined as the ratio of the number of goblet cells to columnar cells in a complete crypt, and the depth of the crypt was the distance from the bottom of the muscular mucosa to the surface of the cavity. A total of 20 crypts were selected for each sample to calculate the ratio of goblet cells to columnar cells.

### 2.6. Intestinal Immune Function and Pro-Inflammatory Factors

Lipopolysaccharide (LPS) in serum was determined with limulus reagent, and the test tube quantitative color limulus kit (EC8045) was purchased from Xiamen Limulus Reagent Biotechnology Co., Ltd. (Xiamen, China). Serum beta-defensin-2 (βD-2) ELISA kit was purchased from Nanjing Jiancheng Bioengineering Institute (Nanjing, China). ELISA for Mucin-2 (MUC-2) level was conducted on intestinal mucus according to the manufacturer’s instructions (E-EL-R0573c, Elabscience Biotechnology Co., Ltd., Wuhan, China). The mixture of PBS added to gut tissues was homogenized on ice using a homogenizer and centrifuged for 15 min at 4000 rpm. The specific ELISA kits were used to evaluate the levels of inflammatory factors in the colon of rats, following the manufacturer’s instructions (Boster Biological Technology Co., Ltd., Wuhan, China). 

### 2.7. DNA Extraction, 16S rRNA Gene Amplification, and Sequencing

One gram of rat feces was collected aseptically and sent to Biomarker Biotechnology Co., Ltd. (Beijing, China). Use Powersoil DNA isolation kit (Mobio, Carlsbad, CA, USA, 12888) to extract bacterial genomic DNA. Common primer pairs (forward primer, 5′-ACTCCTACGGGAGGCAGCA-3′; reverse primer, 5′-GGACTACHVGGGTWTCTAAT-3′) combine the adaptor sequence and barcode sequence to amplify the V3–V4 region of bacterial 16S rRNA gene. Solexa high throughput sequencing technology was used to sequence and analyze the V3–V4 region of intestinal microbiota 16S rDNA. The sequencing platform Illumina HiSeq 2500. FLASH software (Version 1.2.11) was used for splicing the original sequence; Trimmomatic software (Version 0.33) was used for quality filtering of the splicing sequence; and UCHIME software (Version 8.1) was applied to remove chimeras to get the high-quality Tags sequence. UCLUST was used to pick open reference operational taxonomic units (OTUs) at 97% sequence identity. Representative sequences of each OTU were then aligned using PyNAST and assigned based on the SILVA132 database (http://www.arb-silva.de, accessed 13 April 2022). The 16S rDNA gene sequence information was submitted to the Bioproject database under the accession number PRJNA686996 (http://www.ncbi.nlm.nih.gov/bioproject/686996, accessed 13 April 2022). All the biological observation matrix (BIOM) files of each data set were merged using QIIME. The resultant OTU abundance tables were rarefied to an even number (10,000) of sequences per sample to ensure equal sampling depth for the following analysis.

### 2.8. Short-Chain Fatty Acids Measurements of Colon Contents

The colon contents of 9 rats in each group were taken to determine SCFAs. The metabolites in feces were extracted by ultrasonic extraction with 50% sulfuric acid in ice water bath. Seven kinds of SCFAs were quantitatively determined by gas chromatography-mass spectrometry (GC2030-QP2020NX, Shimazu, Kyoto, Japan). The determination was performed on HP-FFAP (30 m × 250 μm × 0.25 μm, Agilent) column with injection volume of 1 μL and flow rate of 1 mL/min. SCFAs were quantified by internal standard method, and standard curves were drawn and quantified by LabSolutions software (Shimazu, Japan). Differential metabolites were searched on the BMKCloud platform (http://www.biocloud.net/, accessed 13 April 2022) and analyzed in combination with intestinal microbes.

### 2.9. Statistical Analysis

IBM Social Science statistical software package (SPSS, Version 25.0, Chicago, IL, USA) was used for statistical analysis, and GraphPad Prim 5 software was used for drawing. The Shapiro–Wilk test and Levene test were used for the normality test and variance homogeneity test, respectively. Normal distribution data were expressed as mean ± standard deviation (SD), and differences between groups were compared by one-way ANOVA with paired comparisons were Bonferroni test (with homogeneity of variance), and when the variance is not uniform, use the non-parametric test method for group comparison and pairwise analysis. The relative abundance of intestinal microbiota did not conform to the normal distribution data, described as the median (25th and 75th percentile), Kruskal–Wallis test for comparison between groups, Dunn test for pairwise comparison with Bonferroni correction. Spearman correlation analysis was used to measure the relationship between intestinal microbiota and SCFAs. *p* < 0.05 was considered to be statistically significant.

## 3. Results

### 3.1. Intestinal Histopathological Analysis

Compared with the CN group, shallow and irregular crypts were found in the HR group (Figure 1A). There was a significant decrease in goblet cell count and ratio of goblet cell vs. columnar cell count in the model group compared to the CN group (*p* < 0.05) (Figure 1D). Interestingly, the number and proportion of goblet cells of rats after *L. casei* intervention increased significantly compared with the HR group, suggesting the protective effect of probiotics on the intestinal epithelium (Figure 1B,C).

### 3.2. Analysis of Related Factors of Intestinal Immunity and Inflammation

Anti-TB drugs led to a reduction of MUC-2 content compared with the CN group, and the intervention of *L**. casei* increased the content than the HR group (*p* < 0.05, Figure 2C), indicating a certain recovery effect of *Lactobacillus casei* to the damage of mucin caused by anti-tuberculosis drugs. Figure 2H showed increased TNF-α levels after H and R intervention. In the meantime, compared with the control group, the supplementary of *Lactobacillus casei* could increase the levels of β-defensin-2, sIgA and IL-10 in colon (*p* < 0.05, Figure 2B,D,F). These results suggested that the probiotic intervention had a beneficial effect on mucus components and immune factors. In addition, there were no differences in the levels of LPS, IL-6, and IL-12P70 among the groups (*p* > 0.05, Figure 2A,E,G).

### 3.3. Analysis on Intestinal Microflora of Anti-TB Drugs and Lactobacillus casei

#### 3.3.1. Alpha Diversity and Beta Diversity Analysis

The study evaluated bacterial diversity in rat feces at week 2 and 6 (Figure 3). Compared with the control group, the OTUs, Chao 1 and Shannon indices in the HR group were reduced by 34.7% (*p* = 0.013), 33.4% (*p* = 0.028) and 24.1% (*p* = 0.016) at two-week, respectively. The number of OTUs and Chao1 index in LLc group and HLc group were lower than these in the control group, and the difference was statistically significant in LLc group (*p* < 0.05), but not in HLc group (*p* > 0.05). Shannon index of LLc group and HLc group were both significantly lower than the CN group (*p* < 0.05). (Figure 3A–C) These results suggested administration of anti-TB drugs significantly reduced the number of OTUs and the alpha diversity indices during the first two weeks, and low-dose *Lactobacillus casei* intervention had no improvement, while high-dose *L**. casei* intervention could only slightly improve the reduction. In the sixth week, the OTUs and Chao1 index have risen and were still significantly lower than control group, but the Shannon index was close to control group (Figure 3A–C). This indicated weaker recovery of OTUs and species abundance and faster recovery of uniformity. Principal coordinate analysis (PCoA) based on unweighted uniFrac distance and PERMANOVA results also revealed that the microbial composition of each group exhibited a distinct cluster at week 2 and 6. The separation mediated by anti-TB drugs could explain the 43.9% and 26.7% changes of the microbial structure (Figure 3D,E). The bacterial structure showed obvious clustering among the CN, HR and two *L. casei* treated groups in the sixth week. Meanwhile, a significant difference was observed between the HR group and the probiotic treatment group (LLc and HLc) (Figure 3E), which was due to the active regulation of intestinal microbes by *L. casei ATCC334*.

#### 3.3.2. The Effect of *L. casei ATCC334* on Microbiome Changes Induced by Anti-TB Drugs

Under the regulation of intestinal microecology, the microbial structure changes induced by anti-TB drugs were recoverable. The microbial structure changed significantly at week 2, and part of the structure recovered at week 6 (Figure 4). Firstly, there was no difference in the abundance of *Lactobacillus* among all groups, whether in week two or week six, but there was an upward trend in *Lactobacillus casei* intervention groups compared with the HR group at week 6 (Figure 4B). Then, we found increased abundance of *Bacteroides* after anti-TB drugs inducing (Figure 4C). Compared with CN group, the abundance of *Akkermansia* and *Blautia* was higher, while *Lacetospiraceae NK4A136 group* and *Rumencoccus UCG-005* was lower in the MOD, LLc and HLc group by the second week. However, there was no difference in the flora abundance between the MOD group and the *L. casei ATCC334* intervention groups (Figure 4D–G). These results indicated that anti-TB drugs significantly altered some bacteria at the genus level, but the probiotics could not restore in a short period of time. At week 6, the use of low doses of *Lactobacillus casei* restored the increase in *Bacteroides* abundance induced by antituberculous drugs (Figure 4C). In addition, there was no difference in the abundance of *Akkermansia* and *Rumencoccus UCG-005* among all groups. By contrast, the *Blautia* and *Lacetospiraceae NK4A136 group* still maintained similar changes to week 2 (Figure 4E,F). However, compared to two-week, there was an overall decrease in *Akkermansia* among all groups, and increase in *Rumencoccus UCG-005* (Figure 4D,G). 

#### 3.3.3. The Effect of *Lactobacillus casei* on SCFAs Changes Induced by Anti-TB Drugs

The quantitative analysis of short-chain fatty acids in feces was conducted at the end of the intervention. Administration of anti-TB drugs reduced the content of butyric acid, valeric acid and hexanoic acid by 73.0% (*p* = 0.003), 46.5% (*p* = 0.009) and 95.6% (*p* = 0.01), respectively (Figure 5A). Compared with the HR group, low-dose and high-dose *L. casei ATCC334* intervention significantly increased hexanoic acid by 20.9% (*p* = 0.028) and 12.1% (*p* = 0.026) (Figure 5B,C). The association analysis between short-chain fatty acids and intestinal microflora clarified that the influence of several microorganisms on the metabolism of short-chain fatty acids (Figure 5D). Interestingly, it was found that three species of bacteria had similar effects on the production of three significantly changed SCFAs. Spearman’s correlation analysis of the different SCFAs between CN and HR groups and the intestinal microbiota after probiotic intervention showed that butyric acid (r = 0.69, *p* = 0.018), valeric acid (r = 0.64, *p* = 0.033) and hexanoic acid (r = 0.68, *p* = 0.022) were significantly and positively correlated with *Lachnospiraceae_NK4A136_group* (Firmicutes). In addition, *Bacteroides* (belongs to Bacteroidetes) was negatively correlated with changes in butyric acid (r = −0.83, *p* = 0.002), valeric acid (r = −0.74, *p* = 0.009), and hexanoic acid (r = −0.77, *p* = 0.005), and *Marvinbryantia* (Firmicutes) was also negatively correlated with changes in butyric acid (r = −0.68, *p* = 0.022), valeric acid (r = −0.69, *p* = 0.018), and hexanoic acid (r = −0.79, *p* = 0.004). In addition, the production of hexanoic acid was affected by a variety of bacteria, in addition to the three mentioned above, there were four species of bacteria was negatively correlated with changes in it. *Candidatus Saccharimo* was positively correlated with the change of hexanoic acid.

#### 3.3.4. The Effect of Anti-TB Drugs and *Lactobacillus casei* on Metabolic Pathways

To investigate the possible mechanism of the certain protective effect of *Lactobacillus casei* on anti-TB drugs induced intestinal injury, the predicted differences in the KEGG (Kyoto Encyclopedia of Genes and Genomes) metabolic pathways were further analyzed (Figure 6). As shown in Figure 6A, the main metabolic pathways of the functional genes in the microbial community were significantly influenced by anti-TB drugs at two-week. These included carbohydrate metabolism, amino acid metabolism, energy metabolism and xenobiotics biodegradation and metabolism. In addition to the metabolic pathways mentioned above, lipid metabolism was also altered after *L. casei ATCC334* intervention (Figure 6B). Although there were significant differences in metabolic pathways at week 2, the only differences in the endocrine system existed between the CN and HR groups at week 6. At the same time, there was no difference in CN and HLc group at week 6.

## 4. Discussion

A six-week animal study was designed to investigate intestinal injury caused by anti-TB drugs and the protective effect of *Lactobacillus casei*. We found that anti-TB drugs damaged the intestinal barriers, including mucus layer composition, intestinal microbiome structure and short-chain fatty acids metabolism, while *L. casei ATCC334* could improve the intestinal barrier function to reduce these damages. This improvement effect might be achieved by raising the number of goblet cells, the content of MUC-2 and the levels of immune factors such as IL-10 and sIgA, increasing the beneficial microbes in the intestines, and improving the metabolism of short-chain fatty acids. 

The mucus system of the colon was attached to the surface of epithelial cells, and MUC-2 secreted by goblet cells was the main structural component [22]. The effect of anti-TB drugs on the intestinal mucus was until now unclear. Our results showed that the intervention of anti-TB drugs reduced the proportion of goblet cells and MUC-2 expression in rat colon tissues. However, this result must be interpreted with caution, because as a broad-spectrum antibiotic, rifampicin might have a direct effect on the mucosa [23,24]. In this study, the intervention of high-dose *Lactobacillus casei* showed a higher proportion of goblet cells than the anti-tuberculosis drugs group. It was speculated that this could be related to the function of probiotics regulating intestinal stem cells to differentiate into goblet cells or goblet cell secretion.

Antibiotics exerted their beneficial effects by killing bacterial pathogens or inhibiting pro-inflammatory mechanisms. However, inflammation might be induced under special conditions, such as early antibiotic exposure for fetuses [25,26], antibiotic abuse [27], induced endotoxemia, etc. Long-term broad-spectrum antibiotic treatment might result in an increase in inflammatory factors (TNF-α and IL-6) [28]. Studies proved that *L. casei* can downregulate the levels of pro-inflammatory factors such as TNF-α, IL-1β and IL-6 [29,30]. In our study, six weeks of anti-tuberculosis drugs had elevated the TNF-α level. There was no significant effect but increase or decrease trend on IL-6, IL-10 and IL-12, which may be related to the dosage of antibiotics and their anti-inflammatory properties. In addition, IgA was an antibody secreted by lamina propria cells of mucous membrane with important immune functions. Recent studies reported IgA that has a symbiotic relationship with intestinal microbes, competes with foreign substances to adhere to the surface of *Bacteroides*, and colonizes on the intestinal mucosa [31], and the decrease of sIgA expression was related to antibiotic-induced immune disorders [23]. It was reported that short-chain fatty acids also appeared to be involved in intestinal immunity [32]. The interaction between changes in gut microbes and microbial regulation of host immunity was very complex. Therefore, we speculated that *L. casei ATCC334* promoted the expression of β-defensin-2, sIgA and IL-10 to protect intestinal from injury, and the effect might be related to the abundance of some intestinal microbes.

Intestinal microbiota colonized the surface of intestinal epithelial cells and had a complex bidirectional relationship with the host [33]. This study demonstrated that anti-TB drugs reduced the diversity of intestinal microbiota, including reducing the number of OTUs and alpha diversity index and altering the taxonomic composition of microbiota. A previous case-control study also supported the effect of long-term anti-TB treatment on reducing gut microbial diversity [34]. Moreover, antibiotics had a rapid and intense effect on intestinal microbiota [35]. Previous study has reported a drastic decrease in diversity after 28 days of treatment with anti-TB drugs [10], but the intervention of our study was 42 days. The powerfully bactericidal effect of anti-TB drugs suggested the changes in intestinal microbes actually predate our observation point (14 days/two weeks). It was demonstrated that TB treatment had dramatic effects on the intestinal microbiome and highlighted unexpected durable consequences of the treatment though part of it was recoverable [10,36], which was similar to what we found in week 6. At the same time, after six weeks of intervention, *Lactobacillus casei* partially restored the structure of the intestinal flora. Firmicutes is the largest dominant phylum in the intestine. Consistent with previous studies, it was also found in our study that the relative levels of Firmicutes were recoverable, with their relative abundance dropped sharply in the second week and returned to a level close to the control group in the sixth week. This recovery might not only do with its own fitness, but also with the benefits of *Lactobacillus casei.* Some changes in microbial structure appeared to be irreversible during anti-TB treatment. The low OTU numbers and low Chao1 index remained unchanged, and some changes in the community structure were also unable to recover, such as the dramatic depletion of *Lachnospiraceae-NK4A136-group* and the increase of *Blautia* and *Bacteroides*. *Bacteroides* was one of the conditional pathogenic bacteria and potentially harmful bacteria with pro-inflammatory effects [1]. Additionally of interest, the changes of *Akkermansia* and *Blautia* in gut have recently received much attention for their probiotic potential. As a dominant genus of intestinal microbiota, Blautia plays certain roles in metabolic diseases, inflammatory diseases, and biotransformation [2]. Compared with healthy individuals, *Blautia* was more abundant in patients with irritable bowel syndrome and ulcerative colitis [37,38], but less abundant in patients with sporadic cancer [39]. It was indicated that the change of *Blautia* was complex, which might be related to the fact that *Blautia* contains many diverse species. These specific changes in *Blautia* abundance may be associated with the multimodulation of microbial metabolism, biotransformation, and autoimmunity. Even if all Blautia-mediated changes probably not be beneficial, it does not mean that *Blautia* played no protective role in the intestine. In addition, due to decreased abundance could be associated with obesity and metabolic disorders, *Akkermansia* has also been in the spotlight, suggesting a beneficial role for this bacterial species [40,41]. Daily administration of *Akkermansia muciniphila* to adult obese mice for 4 to 5 weeks increased ileal goblet cell number and expression of intestinal barrier markers, leading to decreased systemic inflammation and improved metabolic health [40,42]. Our data suggested that anti-TB drugs and/or *Lactobacillus casei* increase the abundance of *Blautia* and *Akkermansia* in rats. Given the protective effects of both species in the gut, this may explain why intestinal damage induced by HR is mild and the protective effects of *Lactobacillus casei*.

In this study, the intervention of *L. case* could not significantly improve the reduction of alpha and beta diversity induced by anti-TB drugs. However, restoration of flora abundance of some intestinal bacterial genus showed protective effects on the intestinal mucosal function and histopathology structure. Although there was no significant difference among all the groups, it was showed that the abundance of *Lactobacillus* was still higher in the LLc and HLc group than the HR group at week 6. Recent studies have suggested that Lactobacillus might modulated the gut microbiota, resulting in increased SCFA producers, such as acetic acid, propionic acid, and butyric acid [43], and was effective at maintaining intestinal epithelial regeneration and homeostasis as well as at repairing intestinal damage after pathological injury [44]. In addition, the abundance of *Blautia* presented the similar trend as *Lactobacillus* between the *L. case**i* intervention groups and the HR group. *Blautia* occupied a dominant position among intestinal microorganisms and produced short-chain fatty acids that could provide energy for colon cells [45], and played an important role in maintaining environmental balance in the intestine and preventing inflammation by upregulating intestinal regulatory T cells and producing SCFAs [2]. The relationship between intestinal microbes and body health was more complicated. Whether this alteration was beneficial remained to be further studied. In addition, the failure of *Lactobacillus casei* to recover all the microbial structure suggested the limited role of single probiotics in regulating gut microbiota.

Short-chain fatty acids are important products of intestinal microbial metabolism and participate in the maintenance of intestinal function. Butyric acid was the main product of intestinal microbes decomposing dietary fiber, providing energy for colon cells, and reducing intestinal inflammation and oxidative stress [46]. Although the use of anti-TB drugs preserved or increased some beneficial bacteria, the levels of butyric acid, valeric acid and hexanoic acid in the colon contents of the anti-TB drug group were significantly reduced. Low-dose and high-dose *L. casei* significantly increased hexanoic acid levels, while the level of butyric acid, one of the major components of colon SCFAs, was only slightly increased. We also found that there was no significant difference between short-chain fatty acids at week 2 and week 6 (Appendix A). It was indicated that continuous *Lactobacillus casei* intervention did not change the content of short-chain fatty acids directly. This might be due to the limited number of beneficial species added by *L. casei ATCC334*. Accordingly, given the changes in microbiota at 2 and 6 weeks, we preferred that the changes in SCFAs were due to changes in gut microbiota. SCFAs acted as endogenous ligands for G-protein-coupled receptors (GPCRs) to exert effects in the organ [47], were related to many physiological functions including involvement in inflammatory responses and regulation of the immune system. In the meantime, butyrate was an essential bacterial metabolite produced in the colon, since it was a preferred energy source for colon epithelial cells, contributing to the maintenance of the gut barrier function, as well as demonstrating immunomodulatory and anti-inflammatory capabilities [48,49]. Anti-tuberculous drugs led to a decrease in butyric acid, which is consistent with the inhibition of the KEGG pathway in energy metabolism, etc. Therefore, we considered that anti-TB drugs might affect the expression of certain pathways by affecting the content of short-chain fatty acids. However, because the limitation of *Lactobacillus casei* promoting beneficial species or number, it may not fully restore the function of related pathway.

This study deeply explored the damage of anti-tuberculosis drugs on various intestinal barriers in rats and found the ameliorative effect of *Lactobacillus casei*. In addition, we tried to elucidate its mechanism from the perspective of metabolites—short-chain fatty acids in our study. However, limitations still existed, including that exactly the mechanisms of intestinal flora or SCFAs responsible for intestinal injury are still unknown. Whether these intestinal changes are related to the metabolic pathway of endocrine system deserves further investigation. *Lactobacillus casei* alone can only sightly recover intestinal damage, suggesting the effectiveness and limitations of *L. casei*. We need explore more trials to mitigate anti-TB drug-induced intestinal injury.

## 5. Conclusions

This study explored the specific aspects of antituberculous drug-induced intestinal injury and the ameliorative effect of *Lactobacillus casei* was effective but mild. Gut microbes and short-chain fatty acids might play important roles in related metabolic pathways. The above results raised concerns about the use of large amounts of antibiotics and the effectiveness of preventive measures. However, anti-TB treatment without anti-TB drugs was impossible. Correct and appropriate use of anti-tuberculosis drugs in anti-tuberculosis treatment, and taking appropriate measures to protect the gastrointestinal tract function was the top priorities. Our study provided some guidance for the search for more effective preventive or therapeutic measures.

## Figures and Tables

**Figure 1 nutrients-14-01668-f001:**
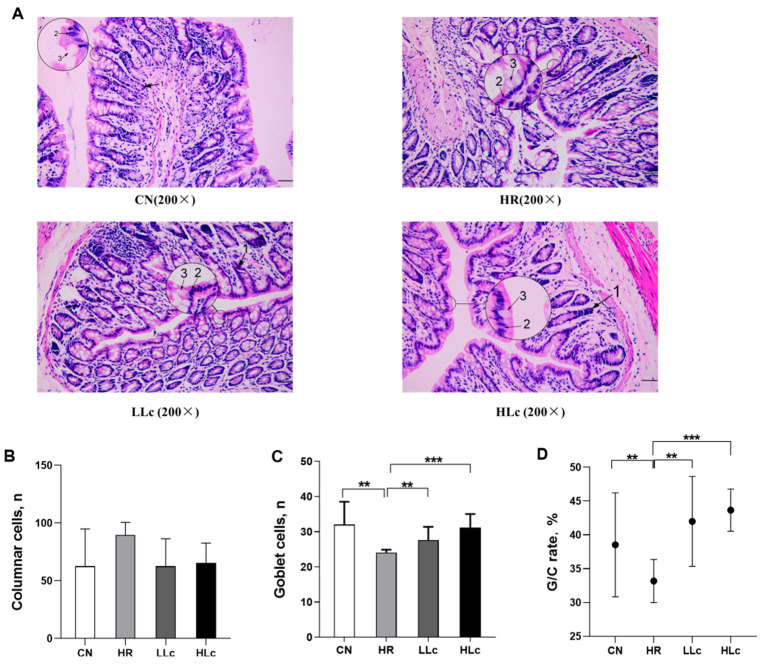
Pathological evaluation and the number of goblet cells of colon tissues of anti-TB drugs and *Lactobacillus casei* for 42 days. (**A**) Pathological conditions were observed under 200-fold field; 1 indicated irregular crypts, 2 indicated columnar cell, 3 indicated goblet cell, Scale bars, 50 μm. (**B**) Columnar cells; (**C**) Goblet cells; (**D**) The proportion of goblet cells to columnar cells. Values are mean ± SD, *n* = 5 per group. CN: control group, HR: Isoniazid + rifampicin model group, LLc: HR + low dose *L. casei ATCC334* group, HLc: HR + high dose *L. casei ATCC334* group. ** indicates *p* < 0.01, *** indicates *p* < 0.001.

**Figure 2 nutrients-14-01668-f002:**
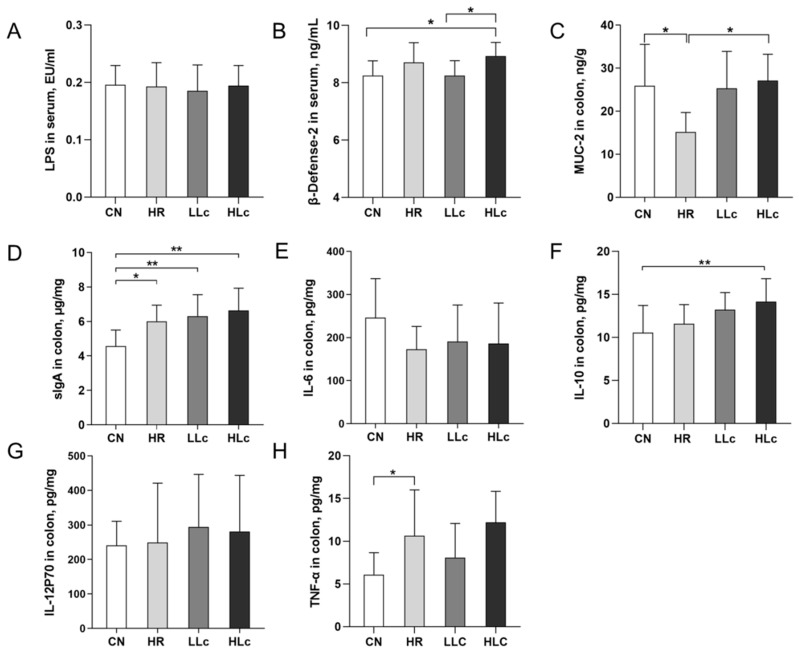
The Effects of anti-TB drugs and *Lactobacillus casei* on colonic immunity and inflammation-related factors. (**A**) LPS in serum, *n* = 10 per group. (**B**) β-defensin-2 in serum, except the CN group *n* = 8, the other groups *n* = 10. (**C**) Colonic mucus MUC-2 in colon, except the LLc group *n* = 8, the other groups *n* = 10. Immune and inflammatory factors in the colon:(**D**) sIgA, (**E**) IL-6, (**F**) IL-10, (**G**) IL-12 p70, (**H**) TNF-α. *n* = 10 per group. Values are mean ± SD. LPS: lipopolysaccharide, MUC-2: Member of the mucin family, sIgA: secretory IgA, IL-6: interleukin-6, IL-10: interleukin-10, IL-12p70: interleukin-12 p70. CN: control group, HR: isoniazid + rifampicin model group, LLc: HR + low dose *L. casei ATCC334* group, HLc: HR + high dose *L. casei ATCC334* group. * indicates *p* < 0.05, ** indicates *p* < 0.01.

**Figure 3 nutrients-14-01668-f003:**
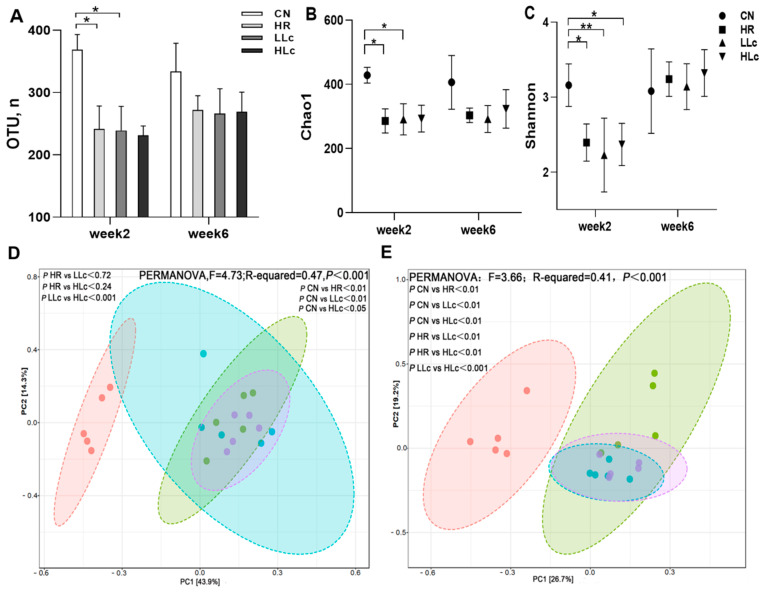
*Lactobacillus casei* improve the reduction of alpha and beta diversity induced by anti-TB drugs. Alpha diversity:(**A**) OUT numbers (**B**) Chao1 index (**C**) Shannon index at two-week and six-week. PCA based on unweighted UniFrac distance to compare the similarity of species types between groups: (**D**) PCoA based on unweighted uniFrac distance on OTU level, week 2; (**E**) PCoA based on unweighted uniFrac distance on OTU level, week 6. Permutational multivariate analysis of variance in weighted UniFrac similarity coefficient (PERMANOVA) was also performed (**D**,**E**). *n* = 5 in each group, values are presented as the mean ± SD, * indicates *p* < 0.05, ** indicates *p* < 0.01. CN: control group, HR: isoniazid + rifampicin model group, LLc: HR + low dose *L. casei* group, HLc: HR + high dose *L. casei* group.

**Figure 4 nutrients-14-01668-f004:**
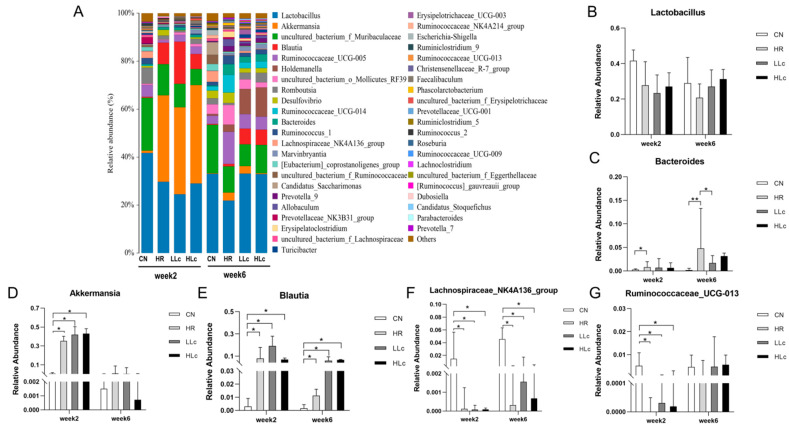
*Lactobacillus casei* and anti-TB drugs change the community structure of gut microbes. (**A**) Relative abundance of microbial taxa determined by 16S rRNA analysis of fecal bacteria at genus level, week 2 and week 6. Percentages of bacteria with greater abundance in the gut microbial communities at the genus level: (**B**) *Lactobacillus*; (**C**) *Bacteroides*; (**D**) *Akkermansia*; (**E**) *Blautia*; (**F**) *Lachnospiraceae_NK4A136_group*; (**G**) *Ruminococcaceae_UCG-013*. *n* = 5 in each group, values are presented as mean ± SD, * indicates *p* < 0.05, ** indicates *p* < 0.01. CN: control group, HR: isoniazid + rifampicin model group, LLc: HR + low dose *L. casei ATCC334* group, HLc: HR + high dose *L. casei ATCC334* group.

**Figure 5 nutrients-14-01668-f005:**
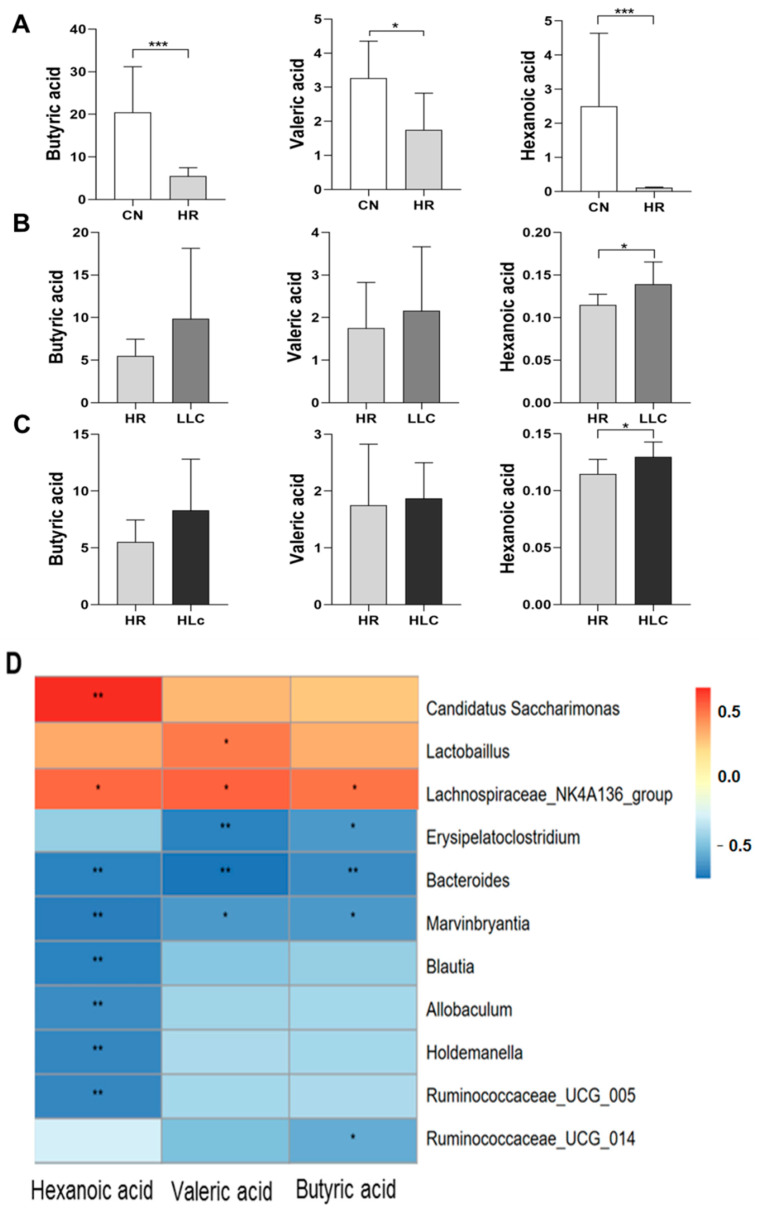
The Effects of *Lactobacillus casei* and Anti-TB drugs on fecal short-chain fatty acids at week 6. (**A**)The effect of anti-TB drugs on short-chain fatty acids in rat feces. The effect of probiotics intervention on short-chain fatty acids in feces of rats with anti-TB drugs: (**B**) Low-dose *L. casei ATCC334*, (**C**) High-dose *L. casei ATCC334.* (**D**) Spearman correlation analysis of short-chain fatty acids and intestinal microbes at the genus level. The top 20 genera in relative abundance were included in the analysis, the correlation coefficient threshold is set to 0.1, *p* < 0.05 was considered statistically significant. * indicates *p* < 0.05, ** indicates *p* < 0.01, *** indicates *p* < 0.001. *n* = 9 in each group in Short-chain fatty acid analysis. The principle of joint analysis of intestinal microbes and short-chain fatty acid content is to match the same rat and the same intervention time. The number of matched rats in the HLc group was 2 and 3 in the other groups. CN: control group, HR: isoniazid + rifampicin model group, LLc: HR + low dose *L. casei ATCC334* group, HLc: HR + high dose *L. casei ATCC334* group.

**Figure 6 nutrients-14-01668-f006:**
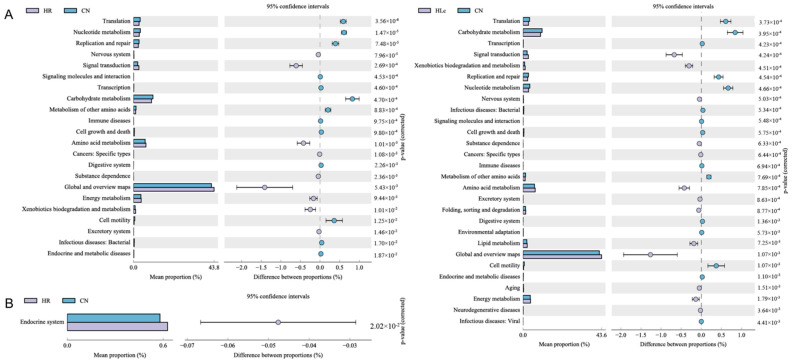
The Effects of *Lactobacillus casei* and anti-TB drugs on KEGG pathways. Imputed metagenomic differences between two groups based on Welch’s *t*-test (*p* < 0.05). The colorful circles represent 95% confidence intervals calculated by Welch’s inverted method. (**A**) week 2; (**B**) week 6. *n* = 5 in each group, CN: control group, HR: isoniazid + rifampicin model group, HLc: HR + high dose *L. casei ATCC334* group.

## Data Availability

No new data were created or analyzed in this study. Data sharing is not applicable to this article.

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
