# Peer review of "Lactobacillus casei Improve Anti-Tuberculosis Drugs-Induced Intestinal Adverse Reactions in Rat by Modulating Gut Microbiota and Short-Chain Fatty Acids"

_nutrients, 2022, doi:10.3390/nu14081668_

Round 1

Reviewer 1 Report

Major points:

  • In Fig.2, authors only measured IL-6, IL-10, and IL-12P70 level in the colons. However, other studies also reported other inflammatory factors, such as TNF-α and IL-1β, also got involved during the development of intestinal injury. Therefore, it’s recommended to add the evaluation of several other pro- and anti-inflammation factors in the colons.

  • As authors suggested, Lactobacillus casei improve the reduction of alpha and beta diversity induced by anti-TB drugs. However, in Fig 3, all the analysis based on samples from 2 or 6 weeks indicated there were still huge differences between CN and LLC/HLC group, which can’t correlate with their findings shown in Fig. 1. A paragraph could be added into the Discussion section for better understanding the relationship between gut microbial diversity and their protective effects in the colons.

  • In Fig 4B-E, authors only made statistical analyses on the differences between CN and HR/LLC/HLC group. Are there any significant differences between HR and LLC/HLC group, which reflected the protective effects of Lactobacillus casei against anti-tuberculosis drug-induced injury?

  • In Fig.5, there are no comparison on the content of valeric acid in rat feces between HR and LLC/HLC group. Does that mean there are no differences?

Reviewer 2 Report

This study explored whether Lactobacillus casei could regulate gut microbiota or short-chain fatty acids to protect intestinal adverse reactions induced by anti-tuberculosis drugs. It is novel to study the anti-adverse effects of antibiotics in microbiota’s metabolic pathways. Hence, I think this study is interesting but can be further improved by addressing some of the following concerns.

Major concerns

  1. What is the minimum inhibitory concentration(MIC) of H and R to L. casei ATCC334. Will the dosage that used for mice already killed the L. casei ATCC334?

     2. Is the L. casei ATCC334 sensitive to pH? If yes, Did the authors gavage            NaHCO3 to neutralize stomach acid before administering L.casei? Did            the authors detect the CFU of L. casei in colon or feces? Just make sure          they are really there.  

  1. Please indicate the irregular crypts, columnar cells, and goblet cells with arrows in figure 1A.
  2. Is that possible to clarify the recovery of short-chain fatty acids is due to the L.casei itself or because of the recovered microbiomes?
  3. Abstract. Line 18-19. “Supplementation of Lactobacillus casei increased the levels of secretory immunoglobulin A and interleukin 10 induced by H and R.” does not make sense. I don’t think the IgA and IL-10 was induced by H and R and L.casei make them more. Please modify the statement based on the data.

Minor concerns

  1. Resolution of figure 6 is too poor.
  2. Figure 1D was missing in result statement, line 193.
  3. Why did the authors only choose the distal colon tissue for histopathology assay? How about other sections and small intestine? Any special reason that distal colon is the most important position for this study?
  4. Discussion about the abundance change of Akkermansia and Blautia could be added to help explain the results and make the article more interesting.
